# Use of Dynamic Shear Rheology to Understand Soy Protein Dispersion Properties

**DOI:** 10.3390/polym14245490

**Published:** 2022-12-15

**Authors:** Charles R. Frihart, Matthew Gargulak

**Affiliations:** 1Forest Products Laboratory, One Gifford Pinchot Drive, Madison, WI 53726, USA; 2AgriChemical Technologies Inc., One Gifford Pinchot Drive, Madison, WI 53726, USA

**Keywords:** soy flour, viscosity, dynamic rheology, shear thinning, dispersion structure, ovalbumin

## Abstract

Soy flour dispersions are used as adhesives for bonding interior wood laminates, but the high viscosity of these dispersions requires low solids in the adhesive formulations; the greater water content causes excessive steam pressure during hot press manufacturing. This limits the utility of soy adhesives in replacing urea–formaldehyde adhesives; thus, understanding the cause of high soy viscosities is important. Lack of literature on aqueous soy flour dispersion rheology led to our dynamic rheology studies of these dispersions to understand high viscosity and the effect of various additives. Even at low soy solids, the elastic nature outweighs the viscous properties at low shear, although increasing the shear results in shear-thinning behavior after the yield point. At even higher shear, beyond the flow point where the storage and loss moduli cross, some of the dispersions show an additional shear thinning transition. The comparison of the rheological properties of aqueous dispersions of the soy flour and protein isolate, and another natural protein, ovalbumin from egg whites, led to a better understanding of different types of rheological behaviors. The experimental observations of two observed shear thinning events for soy are consistent with the model of dispersed particles, forming clusters that then form large scale flocculants.

## 1. Introduction

The large size of the industry for bonded wood products has induced much research on bio-based adhesives [1,2,3,4,5]; however, only oilseeds and lignin are currently available in large enough quantities and can meet the low cost required by this market. Because oilseed flours are the only feedstock that is economically viable for producing light-colored wood adhesives, the Forest Product Laboratory has for years investigated the fundamental properties of soy in conjunction with industry’s applied research to improve soy flour adhesive properties. Among the studies to better understand soy flour performance in wood adhesives, this paper endeavors to develop a fuller understanding of the rheology of these adhesives. The proteins in soy flour are considered the main component for wood bonds with the required dry and wet strength, although the carbohydrates also provide good dry strength.

Protein backbone sequences are difficult to relate to their physical and chemical properties because of the multitude of hydrophobic and hydrophilic domains dictated by the backbone sequences made from varying amounts and order of the 20 monomeric amino acids [6,7]. Of interest as wood adhesives are the storage proteins in oilseed flours (residue after the extraction of the oil and grinding the remaining solid), which are naturally folded into their most compact native structure. Various treatments cause individual proteins to partially unfold and refold into less compact globular structures; any state other than the native state is considered denatured [8]. Although soy flours have been used for commercial wood bonding for nearly a century [9], their properties are not well understood. An important drawback of these adhesives is the rapid increase in viscosity above 20% solids in water as measured using a Brookfield LVTD Digital Viscometer (AMETEK Brookfield, Middleboro, MA, USA), see Figure 1 [9,10]. This high viscosity is problematic since typical competing synthetic wood adhesives are about 50% solids and have reasonable viscosities, meaning that soy adhesives bring comparatively much more water into the bondline, which needs to be removed in the hot bonding process. This excessive water generates greater internal steam pressure that can cause adhesive bond rupture when the press is opened. Consequently, a better understanding of soy protein rheology is an important issue for increased use of these bio-based wood adhesives. The rheology of the purified soy proteins has been extensively studied in relation to food applications [7,11,12], but in these applications the proteins have been modified during the processing, giving them properties unlike the native protein in the commonly used soy flour. While the rotational viscosities using a cup-and-bob assembly are typical laboratory and plant measurements for characterization and process control, they provide limited understanding of the flow behavior of these flour dispersions. The data in Figure 1 shows considerable difference between the native 90 PDI (protein dispersibility index) and the heat denatured 70 and 20 PDI flours, using the Brookfield viscometer, indicating that the history of flour processing is important. The main difference between these flours is the heat exposure in removing the last traces of solvent used in extracting the fatty oil and lipids from the soy, with the native form being exposed to the least heat. On the other hand, dynamic rheology provides much more information, including the separation of the storage (elastic) and loss (flow) moduli over a wide range of shear rates, while the standard laboratory viscometer, such as the Brookfield unit, provides only a single point intrinsic viscosity that is condition and equipment dependent.

The literature is not particularly helpful for several reasons, in that the dynamic rheology of only purified and modified soy protein isolates (SPI) have been studied, and then mainly in relation to gelation for food, fiber, and film applications [13,14,15,16,17]. Soy flour is the only economically viable source of soy protein for wood adhesives, and is composed of only about half protein, with the other half being mainly various carbohydrates with their own contribution to the rheology. In addition, most commercial soy protein isolates that have been traditionally investigated are jet-cooked, with some also being enzyme treated, which greatly alters the protein structure [18]. The emphasis has been on heat gelation to understand the performance in processed foods with little emphasis on the properties prior to gelation [7,19]. Trying to understand the property of each protein fraction has led to the observation that the sum of the parts does not equal the whole SPI, due to interactions between the fractions [7,11,20]. Another limitation is that some studies use dilute solutions or additives for food uses, but adhesives need to be concentrated and with suitable additives for strong bond formation. While foaming is a desirable property for many food applications to provide less dense and more digestible products, it is undesirable in adhesives due to the increased viscosity. Additionally for adhesives, viscosity is lowered by adding sodium metabisulfite (SMBS) to break disulfide bonds, in contrast to the use of enzymes used for viscosity reduction in food applications [7]. Enzymatic depolymerization of soy to lower its viscosity has been found to result in significant strength reductions in adhesive bonding (unpublished observations by Heartland Resource Technology and Forest Product Laboratory researchers). The SMBS has only a minor effect on bond strength, and its use to lower viscosity outweighs this change in small strength in adhesive formulation, which is overcome by the use of a cross-linker.

After developing appropriate rheological analysis methods, the research focused on studying the effect of soy flour concentration, added defoamer, SMBS and cross-linker polyamidoamine epichlorohydrin (PAE) resin, and the differences between native flour, native SPI and commercial SPI. Additional information has been provided by examining the rheology of well-studied systems such as ovalbumin from egg whites and commercial synthetic wood adhesives. This research led to selecting a model for rationalizing the rheology aspects of the soy proteins and that fits with other data on soy. However, other ways of investigating the secondary and higher structures and protein surface properties are needed to reach more definitive conclusions.

## 2. Materials and Methods

### 2.1. Materials

Soy proteins from various sources were used in this study. Commercial soy flours, Prolia™, were obtained from Cargill (Minneapolis, MN, USA) in three varieties: 90 PDI (protein dispersion index [21]), 70 PDI, and 20 PDI. Native state soy protein isolate was produced from the 90 PDI soy flour with the procedure described in Hunt et al. [18]. Insoluble carbohydrates collected as a byproduct of the protein isolation procedure were also used [22] (Note the carbohydrates used for the data presented in this paper were the product of lab-scale protein isolation and therefore were not from the same batch as the pilot plant protein). All sample mixtures included Advantage™ 1529 (Solenis, Wilmington, DE, USA) as a defoamer, unless otherwise indicated. Sodium metabisulfite (Sigma Aldrich CAS 7681-57-4, St. Louis, MO, USA) and PAE (CA1920, Solenis) were used as additives in some cases. Various reference materials were also tested, including ovalbumin purchased from Sigma Aldrich, as well as standard commercial adhesives including Elmer’s Glue-All, Elmer’s School Glue, and Franklin’s Titebond II Premium Wood Glue, purchased at local retail stores.

### 2.2. Sample Preparation

Mixtures of soy flour were prepared by adding sufficient solid material to water to produce a 150 g mixture with the desired wt. % of soy solids, ranging from 10–30%. About 0.1 g of defoamer was added to the water first. Mixing was carried out using a stand mixer with a 2″ disperser blade stirring at 500 rpm while material was being added, followed by a further one hour mixing at 750 rpm. In cases where sodium metabisulfite was used as an additive, a fixed amount of 0.12 g for all mixtures was added along with the flour. In cases where PAE was used as an additive, the volume was calculated so that the solid mass of the PAE matched the set fraction of the solid mass of the soy flour, ranging from 1–10%. The PAE was added to the mixture before the flour and, after mixing, small volumes of 1.5 M NaOH were used to adjust the pH to approximately 6.2, the pH of soy flour adhesives without PAE. For mixtures of soy protein isolate, the total mass was reduced to 75 g in order to save material due to the effort required to obtain this material.

For the insoluble fraction from the purification of the native SPI [18], the material which had been kept frozen without drying was first allowed to thaw. Due to the high water content of this fraction, the mixture components (carbohydrates, water, and defoamer) were combined before mixing, rather than gradually adding the solid components during mixing. Mixing was again carried out at 750 rpm for one hour. After mixing, the pH was adjusted down to 7 using small quantities of 0.1 M HCl.

### 2.3. Rheological Measurements

After mixing, a sample of 25 mL of the mixture was loaded into the rheometer (Anton Paar MCR302 with CC27 cup and bob measuring system, Graz, Austria). The measuring configuration used was a Cylindrical Peltier Thermal Device (C-PTD200) and 27 mm diameter concentric cylinder. After inserting the bob, the sample temperature was equilibrated at 26.7 °C for five minutes. To minimize evaporation at the surface, strips of Parafilm M^®^ were placed across the top of the cup to cover it as much as possible without contacting the stem of the bob. After the temperature stabilized, the instrument carried out a 25-point oscillatory amplitude sweep by ramping the shear strain from 0.01% to 1000% logarithmically at a constant frequency of 1 Hz.

## 3. Results

### 3.1. Rheological Methodology for Soy

Our research, spanning numerous years, has shown that consistent viscosity values for soy adhesives are hard to obtain on a traditional laboratory rotational viscometer without a detailed procedure to control prior shear and heat history and a specified time in the viscometer under shear. This is because similar to most protein dispersions, soy proteins are shear thinning, both in respect to time and shear rate. In contrast, dynamic rheology allows for studies under varying shear conditions, but only the measurements in the linear viscoelastic (LVE) region, where the material behaves in strictly in an elastic manner, can be evaluated by standard rheological theory. The oscillatory LVE is determined by starting at very low shear strains with the shear amplitude being gradually increased past the yield point (YP) at the end of the LVE. Outside this LVE region, additional information is found, such as the flow point (FP) where the storage modulus is equal to the loss modulus, which indicates that the material behavior is changing from solid-like to liquid-like properties.

A dynamic shear rheometer in the oscillatory mode can determine both the storage modulus (G′), which is the in-phase strain response to the applied stress, and the loss modulus (G″), which is the out-of-phase strain response to the applied stress. For a solid-like viscoelastic material, the elastic component (the storage modulus, G′) is greater than the viscous component (the loss modulus, G″) at very low shear strains. An example of the viscoelastic behavior of elastic materials is shown in Figure 2. This example also illustrates that, in addition to determining the solid-like and liquid-like behavior, the dynamic rheology shows the YP when the LVE behavior ends and the FP when the material’s flow is a larger contributor than the elasticity to the overall viscosity. In addition, the data provides the loss factor (LF) showing the solid-like (elastic) nature using aspects of rheology that are well-covered in an applied version [23] and a fundamental version of rheological theory [24].

The use of the common parallel plate method did not work well for our protein because of the wide range of viscosities, shear thinning caused exuding of the samples in some cases, and the cone-and-plate configuration was not viable due to particle sizes being too large. Thus, the cup-and-bob method was used for this soy rheology work.

The first step was to make sure that rheological measurements started out in the LVE where standard rheology theories apply. This is performed by starting at very low shear strain in the oscillatory mode, and then increasing the amplitude of the oscillation until the final set point is reached. This is important because the soy flour has a linear behavior up to the yield point when shear thinning begins. The shear thinning is important for better mixing and flow through the applicator of “high viscosity” soy dispersions.

This research focused on mainly the flour of 90 protein dispersibility index (PDI) since this flour is considered the native structure due to limited heating during processing. Of course, the soy adhesives are more than just soy flour in water; thus, the effect of additives also needed to be investigated. A defoamer is used to minimize the air from both the solid soy flour and the wetting out process from being incorporated into the dispersion. The dispersion density is lowered with an increasing amount of air incorporation, and prior Brookfield viscosity work showed an undesired increase in viscosity as the amount of air incorporated increased. Since defoamers decrease surface tension, there is the potential for defoamer to influence the measured soy rheology. In Figure 3, the data show that the effect of adding defoamer was small in that it did not change the general shape of the curve from low (10%) to high (25%) soy flour dispersions, but did slightly alter the absolute moduli values. Thus, the defoamer was used in all subsequent tests to limit the effect of aeration on the soy rheological properties. As expected, the soy dispersions show significant solid-like behavior prior to shear thinning that increases with higher solids due to greater protein structure development. The data also show a second unexpected shear thinning event, with one before and the other after the flow point, and these two events are clearer at the higher concentrations. The literature is not clear about these events and explanations will be discussed later in the paper.

In addition to the G′ and G″, the rheological data allow the calculation of the complex viscosity. As shown in Figure 4, the initial complex viscosity (G*/omega), where the complex modulus, G* is the stress amplitude divided by strain amplitude [23,24] decreased rapidly with mixing time until it levelled out at about 60 min of mixing at 750 rpm, showing that the soy has time-dependent character as well as shear-dependent character. This viscosity drop may be due to the loss of air entrapped by the solids during the wetting out process and/or the soy dispersion structure equilibration in water. Some of the earlier experiments were conducted using 60 min mixing time and others at 120 min in later experiments, but little difference in the important parameters was observed between the two times in repeated experiments.

### 3.2. Analysis of Soy Flour

Having established the procedure for preparing the samples for dynamic rheology, we determined the effect of the soy flour concentration of the commonly used 90 PDI soy flour on the rheological curves. Under the experimental conditions, the values of the LVE region, the YP, and FP can be determined. Due to the magnitude values, the data are plotted on log–log scales with amplitude change being plotted as the more intuitively meaningful shear stress value, rather than the applied amplitude change.

As expected, Figure 5 shows that the highest G′, G″, yield point and flow point are with the 30% soy flour solids, and each of the values decreases with decreasing concentration. The good reproducibility for each concentration is shown in Figure A1, Figure A2, Figure A3, Figure A4 and Figure A5 in the Appendix A by including the data for each of the three samples run at 15, 20, 25, and 30% solids. Even at the 30% soy solids, the flow data starts to dominate at about 2 Pa, showing the flour dispersions should have good flow properties with minimal applied shear, especially with the large fall off in the elastic component. At higher shear amplitudes, an unusual second transition is observed in both the G′ and G″ curves, suggesting there may be two levels of structure for the flour. What these transitions may tell us about the soy structure will be discussed later in the paper after additional data is presented.

Figure 6 provides another way of examining the data by plotting the loss factor, which is the G″ divided by G′ to show the elastic character of the sample that is not as easy to visualize in Figure 5. Loss factor values less than 1 have predominantly an elastic character, while those greater than 1 have predominantly flow characteristics. The data in Figure 6 more clearly shows that with increasing solids content, the elastic nature of the dispersion becomes a more significant factor. However, there is still considerable interaction of the soy particles at the 10% solids level, as illustrated by the shear thinning, and the force needed to disrupt the particle–particle interaction. It is clear from Figure 5 and Figure 6 that the soy dispersions are non-Newtonian materials. The initial drop in the storage modulus represents the fracture of the large-scale flocculated structure. The second transition at higher shear forces affects the G′ much more the G″, as shown by the large decrease in the storage modulus in Figure 5. This is likely due an additional structural element that takes more shear force to completely disrupt the flour structure, but does not have much influence on the flow properties. Given the globular nature of the individual protein chains, this second transition is probably breaking the protein aggregates caused by the surface hydrophobic attraction and surface polar bonds of the protein globules.

Proteins receive the most attention as soy flour adhesives because they are the main adhesive component for dry and wet strength, but they are only about half of the flour composition. The other half consists of equal portions of soluble and insoluble carbohydrates. These are isolated in the protein purification process in which the proteins are separated from the insolubles (higher molecular weight carbohydrates and insoluble proteins) by centrifugation of an aqueous 90 PDI dispersion at a pH of about 8, and then collected as solids after lowering the pH to about 4.5 to separate it from the solubles (lower molecular weight carbohydrates, soluble proteins, and salts). The solubles should not influence significantly the rheology since they are mainly low molecular weight sugars, such as sucrose, raffinose, and stachyose, and low molecular weight proteins [25]. On the other hand, the insolubles are made up of polysaccharides that can be separated into major non-cellulosic and minor cellulosic internal cell wall structural parts. The non-cellulosic portion is mainly the acidic polysaccharide D-galacturonic acid and L-rhamnose chains with side chains consisting of mainly galactose and arabinose residues. As well as the carbohydrates, this insoluble fraction also contains some high molecular weight proteins. Consequently, we considered that this insoluble fraction could contribute to the overall soy flour rheology profile. In Figure 7 with the replicates in A6, it is clear that they do contribute and show a very large drop in the storage modulus once the matrix begins to fracture, but the proteins with their greater weight fraction contribution are the dominant factor in flour behavior, as shown in this figure of the purified native soy protein isolate (PPSPI), which showed only a single transition. The two transitions were not limited to the native 90 PDI flour, but were also observed with the partially denatured 70 PDI flour. With neither the insoluble nor the purified protein fractions showing a second shear thinning transition, a possible explanation is that the first transitions may be due to protein–carbohydrate interactions [26]. All the samples were subjected to the same change in amplitude conditions; the difference in the sample shear resistance resulted in different starting and stopping points when plotted as shear stress curves.

Although the native 90 PDI flour can be used for making wood adhesives, there are two other common soy flours, the denatured 20 PDI and 70 PDI. Figure 1 shows that they had higher viscosities than 90 PDI flour in a Brookfield rotational rheometer at a higher concentration, but this does not tell us much about the dispersion properties and whether they have the same types of curves whether the flour is denatured or not. Thus, the information provided in Figure 8 and the replicates in A7 shows some similarities in the flow point and having two transitions, but in the low shear region, 70 and 20 PDI curves have higher G′, G″, YP, and elasticity than the 90 PDI flour. However, the drop in the G′ is steeper for the 70 and 20 PDI in the first shear thinning segment. After the FP, the 20, 70 and 90 PDI are similar in behavior, which is easier to see on the plot of the loss factor versus shear stress in Figure 9. Significantly, the 20 PDI, the most denatured protein variety, showed the highest elasticity at the low shear rate. This may not be surprising because a low PDI indicates that the proteins have low solubility in water (PDIs are determined by the percentage of original nitrogen content left in the supernatant after mixing and centrifugation [21]), which may result in the high elastic interaction for the particles in the 20 PDI flour. The difference in the soy flours before the FP is emphasized by plotting the loss factor for the three different PDIs in Figure 9 with the replicates in A8.

### 3.3. Effect of Additives

As discussed previously, there are multiple drawbacks with using soy flour in adhesive applications. In addition to the wet strength of the bond not being as high as required [10,27,28], the viscosity of soy dispersions is high enough to limit the solids content of the adhesives. The high molecular weight of the protein agglomerates (180 to 300 kDa) contributes to the high dispersion viscosity, but as discussed later in this paper, the protein-protein and protein–carbohydrate interactions also play a very important role. In food applications such as protein drinks, the viscosity of the commercial SPI dispersions (CSPIs) can be greatly reduced by enzymatic cleavage of the proteins [7]. Typically, cutting polymer chains seems detrimental to providing improved strength, although with proteins one cannot predict how chain alteration influences the interactions between protein globules. Mo et al. showed that even very specific 11-unit peptides can provide wood bond strength [29]. However, Frihart and Birkeland were able to verify that CSPIs using hydrolyzed protein had lower wet strengths along with the lower viscosity [30]. CSPIs are not the same as the native SPIs, since the former are jet-cooked versions of the later to increase their performance functionality; this also greatly increases their viscosity [30]. Another way to lower the viscosity of soy dispersions is to use a pH around 4.5, the isoelectric point, but this is where the proteins precipitate and hinders the formation of stable dispersions and may alter protein or cross-linker reactivity.

Another route to lower aggregate size is to break the disulfide bonds by adding thiols or sodium metabisulfite (SMBS) [7,31], with the latter favored for adhesives since it does not impart the thiol odor to the product or require a basic pH. Instead of shortening the protein chain as occurs with enzymatic processes, the SMBS breaks the side chain disulfide bonds between chains, which is well known to disrupt the glycinin structure [7,31]. This also greatly reduces the soy dispersion viscosity as measured on a Brookfield viscometer. Because the use of SMBS is an effective way to reduce soy dispersion viscosity and is commercially viable in making wood adhesives its effect on soy dispersions was investigated. The curves in Figure 10 and the replicates in A9 show the effect of SMBS on mixtures of 30% 90 PDI flour, with reductions of the G′, G″, YP and FP. The elasticity (G′-G″) in the LVE region increased, but the strong transition in G″ and G′ at higher shear stress beyond the flow point is still present.

The effect of the SMBS addition becomes more important at lower soy flour solids as shown in Figure 11. While in all cases the G″ decreases, the G′ decreases to a greater extent. As a result, the 15% soy solids did not have a stable LVE for the G′ under these tests conditions. Since the SMBS does not alter the backbone amino acid sequence or lower the individual protein molecular weights and should have little effect on the overall hydrophilicity and hydrophobicity of the protein, the viscosity change is most likely a result of a change in protein aggregation due to changes in the protein surfaces. From a practical standpoint, the higher soy solids are the ones used commercially; thus, emphasis should be placed on those results in Figure 11.

Currently, most commercial soy adhesives for wood bonding use polyamidoamine-epichlorohydrin (PAE) resin to cross-link the soy particles [9,32,33]. The PAE–soy has sufficient stability to examine its rheological properties under the current testing conditions but increases in temperature or pH cause the viscosity to increase more rapidly due to reaction of PAE with the soy and itself. It is interesting to note that there is virtually no change in the rheological curves up to 2.5 wt% PAE based on the soy weight, but higher amounts cause an increase in the G′, G″, and YP, see Figure 12. The aforementioned second transition after the flow point is present and maybe more distinct. It is clear that the PAE has a large effect on the rheology at higher amounts in moving the yield point of the dispersion to higher shear forces and showing more solid character at low shear levels and showed a steeper change in the first transition, as shown in Figure 12. Figure 13 shows that more PAE leads to a more solid like behavior indicating strong interaction of PAE and protein. Zhong et al. examined the complexation of native soy protein isolate and PAE and found the interaction peaked at 5 wt% soy [33]; the interactions were demonstrated using ultraviolet and infrared spectroscopy, and conductivity titration as a function of pH. The interaction of the cationic PAE with the negatively charged proteins is not surprising, but the increased structure with higher amounts of PAE is shows a strong interaction between the flour components and the PAE. The second transition was still present, but showed a steeper drop in G″. To be sure that this effect is not due to the PAE component, the rheology of the PAE was determined under similar conditions using 20% PAE solids. Figure 14 shows that the PAE behaves as a Newtonian liquid and therefore the unusual behavior exhibited in Figure 12 and Figure 13 is due to protein–PAE interaction.

### 3.4. Non-Soy Adhesives

Because the flour and soy protein isolate are mixtures of very different protein structures [7,11,34], ovalbumin (OV) seemed like a simpler model compound for studying protein rheology than native SPI. The OV with a high protein content can be readily obtained from egg whites and is commercially available in its native state, compared to the multi-step process needed for the SPI isolation. Later it was realized that the fallacy in this logic is that OV is nothing like native SPI with the former being a monomeric, phosphoglycosylated protein, and the latter being a mixture of multi-unit glycosylated proteins. However, it is useful to compare differences in proteins from different natural sources. From the curves in Figure 15, the OV (15% protein) is a fairly Newtonian liquid, while the two soy materials have significant elastic character. Even at a high solids content of 40% OV, the proteins show limited elastic interaction despite the steric crowding due to limited free volume. High flow resistance is illustrated in Figure 16 of the loss factor versus shear stress, which shows that the loss factor is always above 1, indicating low elasticity even at 40% concentration, but greater resistance to flow, which shows the lack of a steric crowding effect when little protein–protein interaction is present. These data reinforce the idea that the high viscosity properties of soy are due to stronger protein–protein and protein–carbohydrate interactions than protein–solvent interaction.

For understanding soy adhesive rheological behavior compared to some typical adhesives, the amplitude shear stress curves are shown in Figure 17. Elmer’s Glue-All and School Glue and Franklin Titebond II show a fairly long LVE with a G″ higher than the G′, despite the about 50% solids contents, although they show changes in G′ and G″ at higher shear rates, especially the School Glue. The cause of this change is unclear, but it could be due to the distortion of the dispersed particles to become more oval in shape to align with the shear force [23,24]. This reduces the viscous drag and can influence inter-particle interaction. The data show that soy behaves very differently from other adhesives in being more elastic at low shear rates and showing high inter-particle interaction, while the others show low inter-particle interaction.

## 4. Discussion

Soy proteins behave very differently than typical wood glues and this needs to be considered in applications, because a simple rotational viscosity measurement initially shows that the proteins have much higher viscosity than the phenolic, amino, polyurethane, or polyvinyl acetate adhesives. However, the mixing and application to wood involve high shear rates, which causes the protein adhesives to flow readily due to shear thinning, unlike other wood adhesives. This naturally leads to the question as to why the soy flours and their components are shear thinning. To understand this issue, it is valuable to understand what the oscillatory shear experiments are actually measuring. The predominant phase is water, but there must be high association between molecules in the soy flour to produce an elastic response. Newtonian behavior is true of most polymer dispersions, including most wood adhesives, where the dispersed particles interact more with the aqueous phase rather than each other. Deviation from this pattern indicates some other phenomena exists, such as deformation of the particle shape from spherical to elliptical to allow less resistance to the oscillating water flow past the dispersed particle. However, with protein dispersions, the particles have significant interaction with each other [7,12,35]; thus, for fluid flow, sufficient force is needed to break these attractions. Therefore, if the attractions are weak, the yield point is at low shear forces, but with higher inter-particle attraction forces, more external force is needed to allow free fluid flow. In contrast, the monomeric OV has stronger interaction with the water phase than with other proteins, is liquid like, and is not shear thinning.

Proteins are complex structures with hydrophobic, non-ionic hydrophilic, cationic, and anionic groups. At the isoelectric point (pI), the cationic and anionic charges are balanced, and the protein particles have high hydrophobic interaction with each other and often precipitate. For soy dispersions, most of the proteins have a pI around 4.5; at this point, these proteins precipitate and the viscosity is minimal. Away from the pI, inter-particle electrostatic force becomes important in relation to the hydrophobic forces, causing the protein particle to disperse by better interaction with the aqueous phase. However, there is still a strong hydrophobic interaction between soy proteins [12]. The soy rheology behaves as a visco-elastic gel, which shows solid-like behavior prior to yielding.

Only when the pH is above 10 are the repulsive forces great enough to overcome the hydrophobic attraction. This knowledge led to development of protein adhesives in the early 20th century, when proteins were dispersed at high pH using monovalent bases that were then displaced by polyvalent cations as cross-linkers during the bonding process [36] which allowed the adhesive to develop the strength needed for good bond formation. However, this process does not make a product that meets the current performance requirements. In current applications, where PAE is used as a cross-linker, high pH conditions are not viable due to PAE self-reaction. In addition, the lower viscosities of high pH conditions have poor stability over time, meaning that the short pot-life limits production schedules.

In the case of ovalbumin, the electrostatic repulsion keeps the hydrophobic domains buried and there is little interaction between individual proteins [37]. Thus, the lack of enough hydrophobic groups on the surface allows the protein to be monomeric, in contrast to most proteins. The gelling of these egg proteins has been explained as α-to-β structural transformation of ovalbumin to expose hydrophobic groups that form strong water-resistant bonds [38]. This is a logical explanation for the data in Figure 14 showing that the ovalbumin is liquid-like having a loss modulus much higher than the storage modulus until the OV gels under heating.

However, soy proteins are known to form aggregates, such as glycinin and conglycinin, under dilute conditions [7,11,39]. Ongoing unpublished research at the FPL has shown that particle size measurement for the PPSPI is about 350 nm while the crystal size of even the dominant glycinin is about 10 nm. This an indication that even larger agglomerates exist than are normally considered for soy proteins in the quaternary structure. Protein literature has discussed protein clusters being in a pre-gelled state as shown in Figure 18 [19,35,40], which may explain the large particle sizes. The surface hydrophobicity of the soy protein is likely the driving force to form these clusters. In addition, these proteins are known to have both strongly bonded non-freezing and freezing water layers around them [7]. Both the swelling of the protein by the water layers, in addition to the cluster formation, would make the soy proteins take up a larger volume than would be expected by the crystalline protein structures. In addition, the clusters are less likely to be spherical, which could also increase the viscosity. The shear thinning transitions may be an indication of the energy needed to break the long-range interaction of these clusters and flocks. Depending on the solids concentration, the YP can be from 4 for flour up to 40 Pa for the native PPSPI.

However, at this point, there is not a clear explanation in the literature of why soy flour has two shear thinning transitions. One possibility is that there are larger flocs of the clusters that are more easily broken, and the second transition is breaking up of the clusters (Figure 18). The larger flocs can involve the carbohydrates as well as the proteins since the dynamic rheology curve of the flour is different than either that of the SPI or the insoluble fraction, as illustrated in Figure 7.

## 5. Conclusions

There is abundant literature of the soy protein isolate rheological gelling properties, especially the denatured jet-cooked version because of its importance to food products [7,15,16,41,42]. This includes rheology that is important not only to the mixing and processability in making the final food products, but also to the products’ texture, taste, and feel. On the other hand, the defatted meal is used for animal feed or for fermentation into food products. Although soy flour rheology is also important for its use as a wood adhesive, detailed studies have not been reported to date. In our rheological studies of amplitude sweep, G′ > G″ in the LVE range shows behavior similar to a visco-elastic gel, in that the soy shows solid-like behavior prior to yielding. While the native soy protein isolate has a typical single shear thinning transition, the native 90 PDI flour has two shear thinning transitions, even after allowing for the time-dependent shear thinning during dispersion preparation.

The effect of adding defoamer to limit the air incorporation, which reduces dispersion density and viscosity increase, is minimal. However, adding sodium metabisulfite to cleave the disulfide linkages between protein chains had a much larger reduction in the storage and loss moduli although it did not eliminate the two transitions. In contrast, the rheology of ovalbumin from egg whites, with its surface hydrophilic groups dominating its hydrophobic tendencies, is similar to that observed for typical synthetic polymeric adhesive dispersions, which are visco-elastic fluids. In addition, high levels of ovalbumin become thicker, showing that steric crowding is not the reason that soy dispersions are shear thinning. While typical organic polymers behave similar to cooked spaghetti, forming entangled structures, the soy proteins are globules with their strong protein–protein interaction similar to sticky rice and the ovalbumin with its low protein–protein interaction behaves similar to non-sticky rice. The rheological properties of soy flour dispersions can be explained by formation of pre-gel clusters and flocs, with the loosely bound flocs yielding first, followed by breaking up of the clusters at higher shear forces (Figure 18).

## Figures and Tables

**Figure 1 polymers-14-05490-f001:**
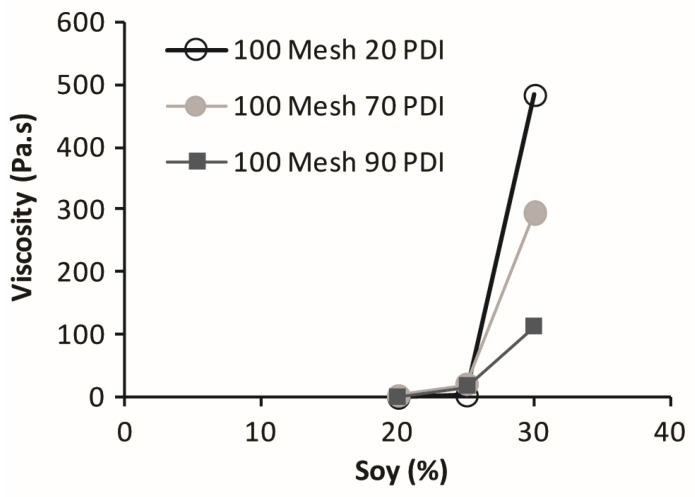
Effect of soy solids content and protein dispersibility index (PDI) on viscosity of adhesives made from 0.152mm (100 mesh) soy flour, data from [10].

**Figure 2 polymers-14-05490-f002:**
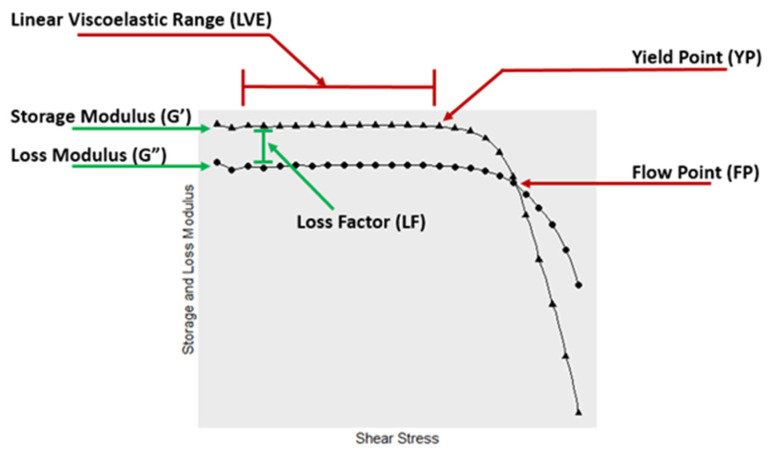
Standard log–log plot of a viscoelastic material where G′ is the storage modulus and G″ is the loss modulus, and both are plotted against shear stress.

**Figure 3 polymers-14-05490-f003:**
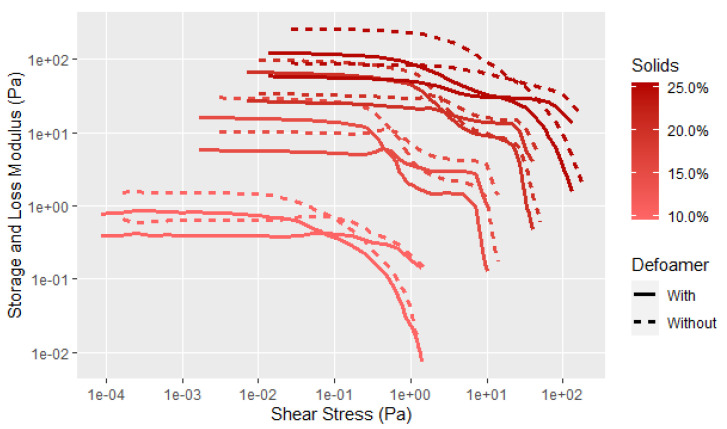
Comparison of the effect of a defoamer from low (10%) to high (25%) soy flour dispersions where the mixing time prior to rheological measurement was 15 min. The data at low shear stress, as in the lower solids, is more variable at the lower measurement limit.

**Figure 4 polymers-14-05490-f004:**
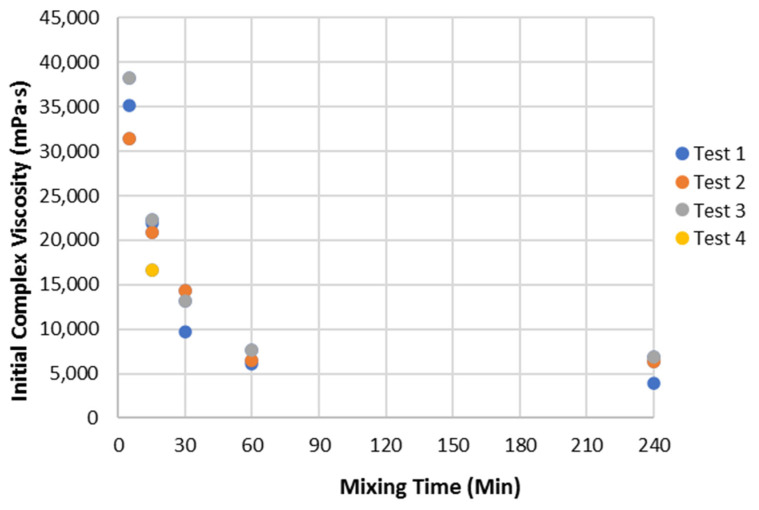
Effect of mixing time on the complex viscosity in the LVE range for 25% solid mixtures of 90 PDI soy flour in water.

**Figure 5 polymers-14-05490-f005:**
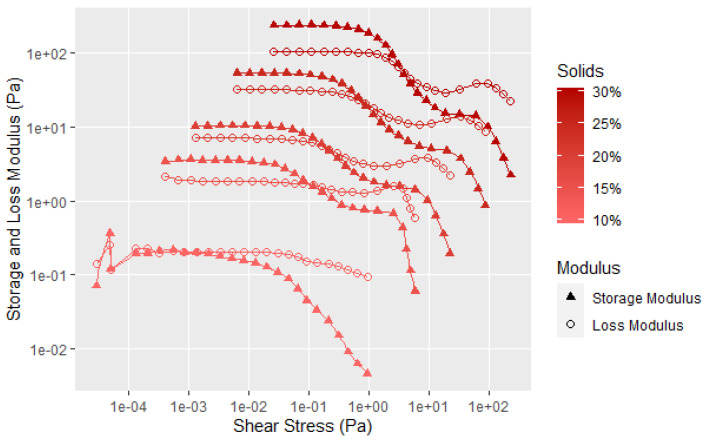
Averaged (3 replicates) amplitude sweep data for mixtures of 90 PDI flour with 1 h mixing time and varying solids content. Individual runs are shown in Figure A1, Figure A2, Figure A3, Figure A4 and Figure A5 (Appendix A).

**Figure 6 polymers-14-05490-f006:**
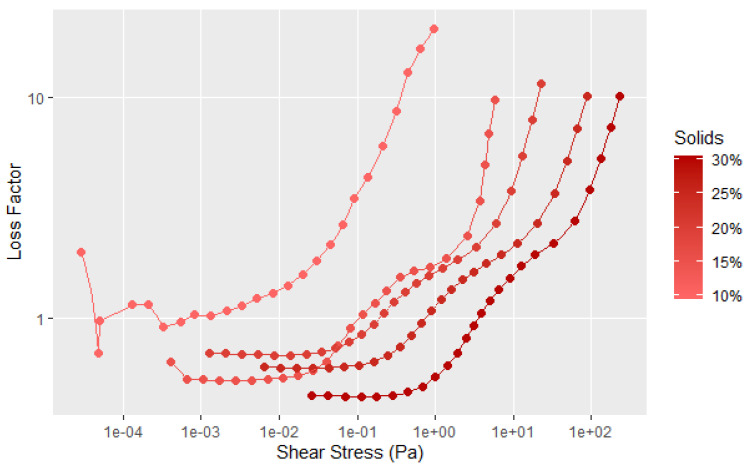
Loss factor data for tests shown in Figure 5, with the lines showing the average of 3 to 4 runs.

**Figure 7 polymers-14-05490-f007:**
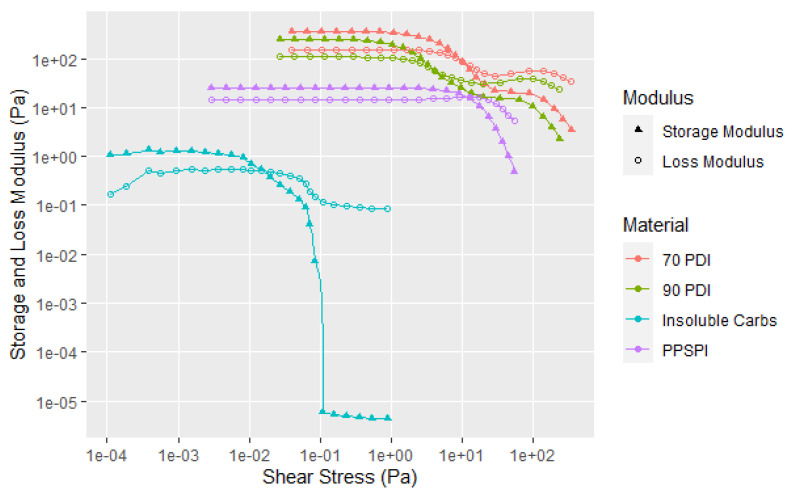
Storage and loss moduli versus applied stress for two soy flours (30 wt. %), native soy protein isolate and soy insoluble fraction (at 15 and 7.5 wt. %, respectively, representing their proportional contribution to the soy flour). Single replicates shown (not average). All replicates shown in A6.

**Figure 8 polymers-14-05490-f008:**
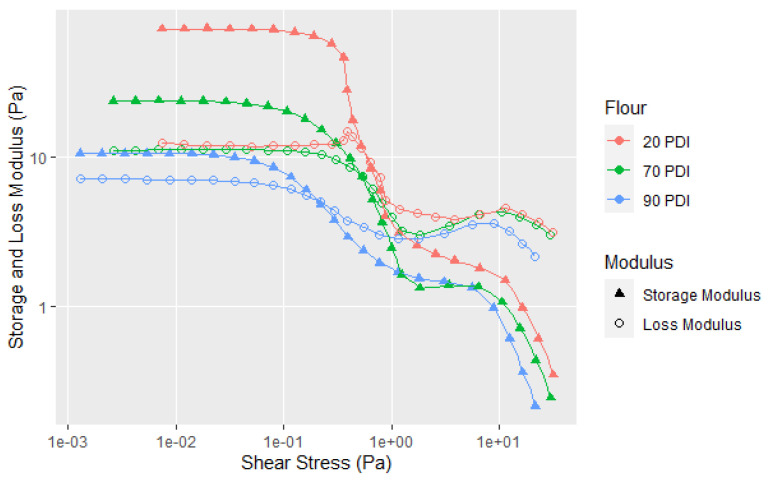
Amplitude sweep data for 20, 70, and 90 PDI flour at 30% solids. Single replicates shown (not average). All replicates shown in A7.

**Figure 9 polymers-14-05490-f009:**
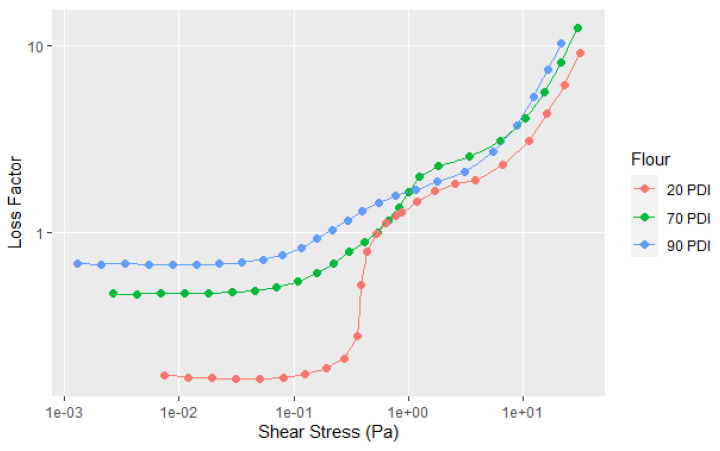
Loss factor data for the samples shown in Figure 8. Single test shown (not average). All replicates shown in A8.

**Figure 10 polymers-14-05490-f010:**
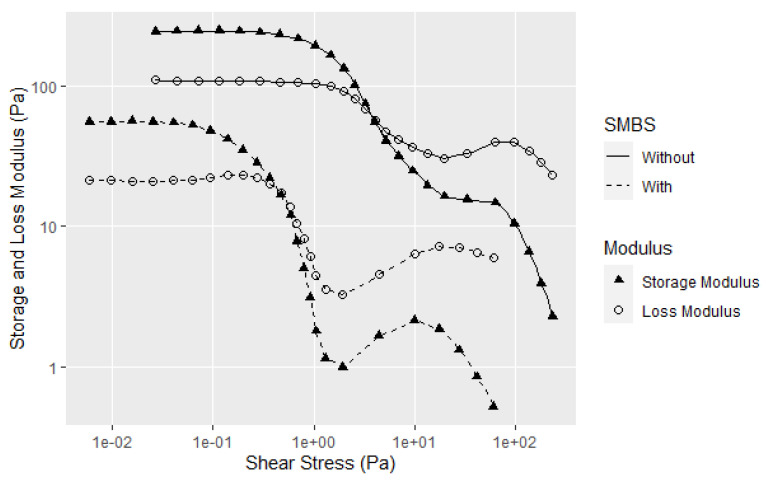
Amplitude sweep data for samples of 30% 90 PDI flour with 60 min of mixing, before and after the addition of SMBS. Single test shown (not average). All replicates shown in A9.

**Figure 11 polymers-14-05490-f011:**
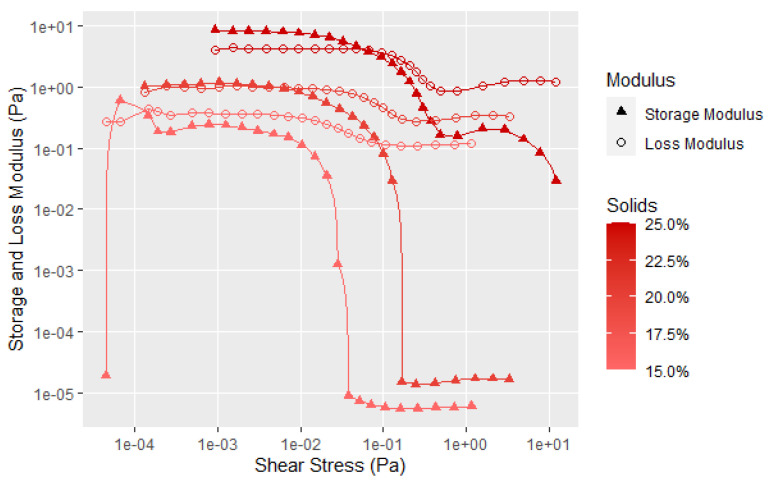
Amplitude sweep data for samples of 15, 20 and 25% 90 PDI flour with 60 min of mixing after the addition of SMBS. For the 15% sample the first data points are more variable under these operating conditions. Single test shown (not average). All replicates shown in A10.

**Figure 12 polymers-14-05490-f012:**
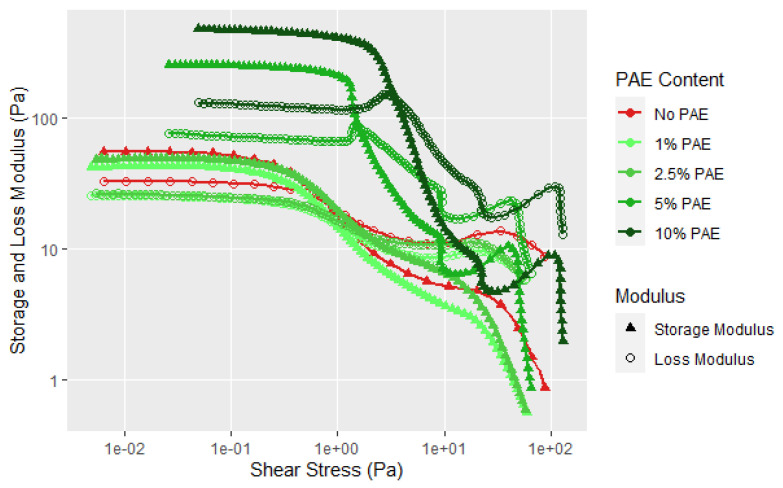
Shear rheology with increasing amplitude for soy flour (90 PDI at 25% solids) without and with the addition of PAE at solids levels relative to soy solids.

**Figure 13 polymers-14-05490-f013:**
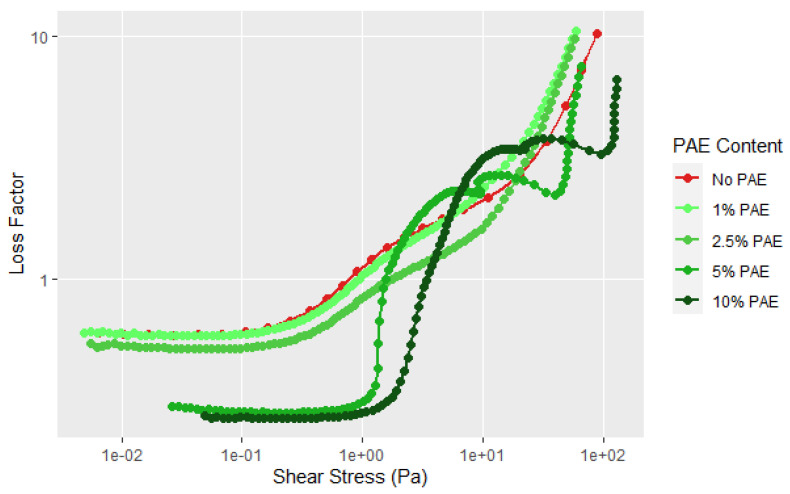
Loss factor curves for the data given in Figure 11.

**Figure 14 polymers-14-05490-f014:**
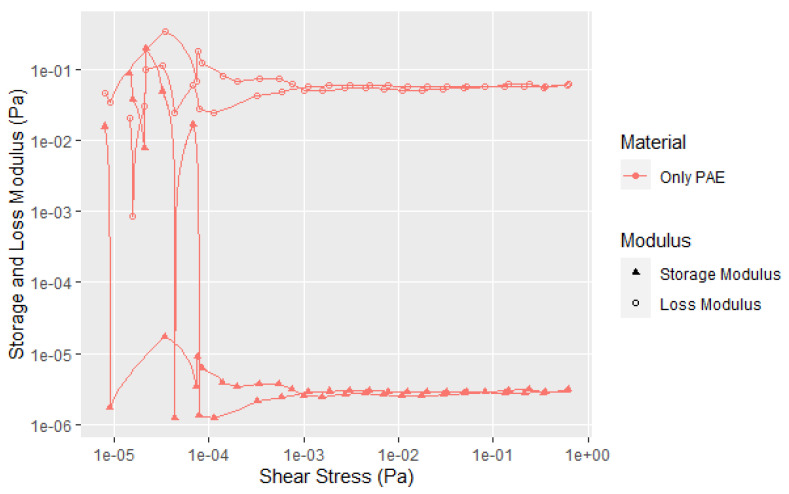
Shear curves for solutions of PAE at the same concentration as the “5% PAE” mixtures shown in Figure 12 and Figure 13 and with pH adjusted to pH 7. For consistency, the PAE was run under the same conditions, even though the first few data points are at the machine limits of consistent measurements.

**Figure 15 polymers-14-05490-f015:**
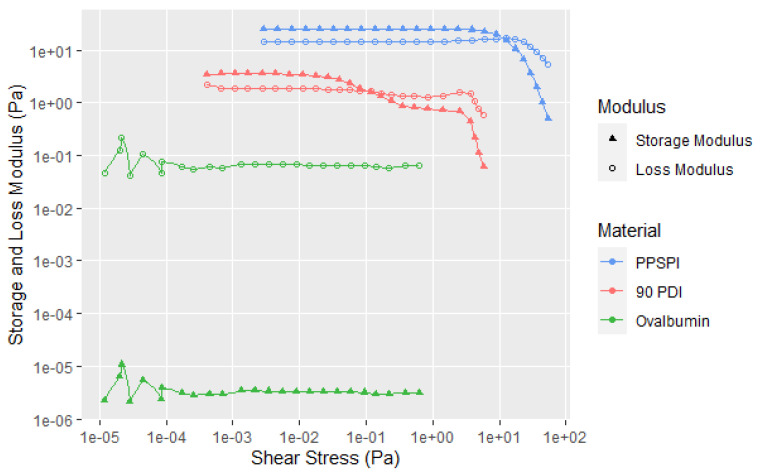
Shear rheology with increasing amplitude of PPSPI, 90 PDI flour, and ovalbumin, all with 15% protein content.

**Figure 16 polymers-14-05490-f016:**
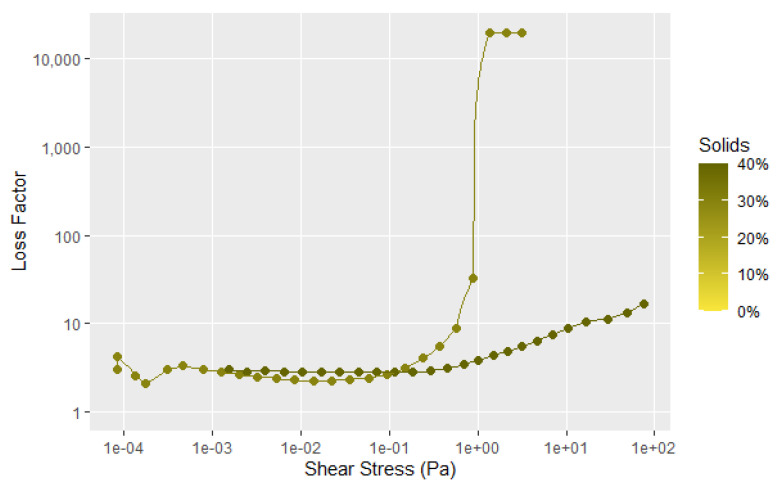
Loss factor for shear stress rheology with increasing amplitude testing of OV with 30% and 40% concentrations.

**Figure 17 polymers-14-05490-f017:**
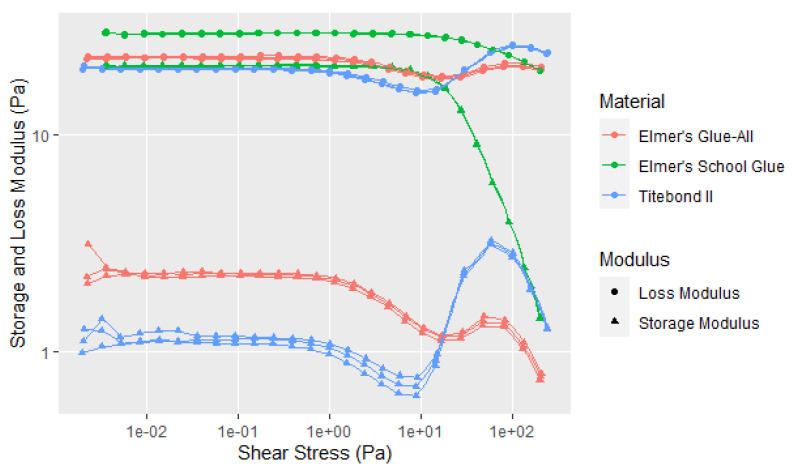
Shear rheology with increasing amplitude of commercial PVA-based adhesives: Elmer’s Glue-All and School Glue, and Titebond II.

**Figure 18 polymers-14-05490-f018:**
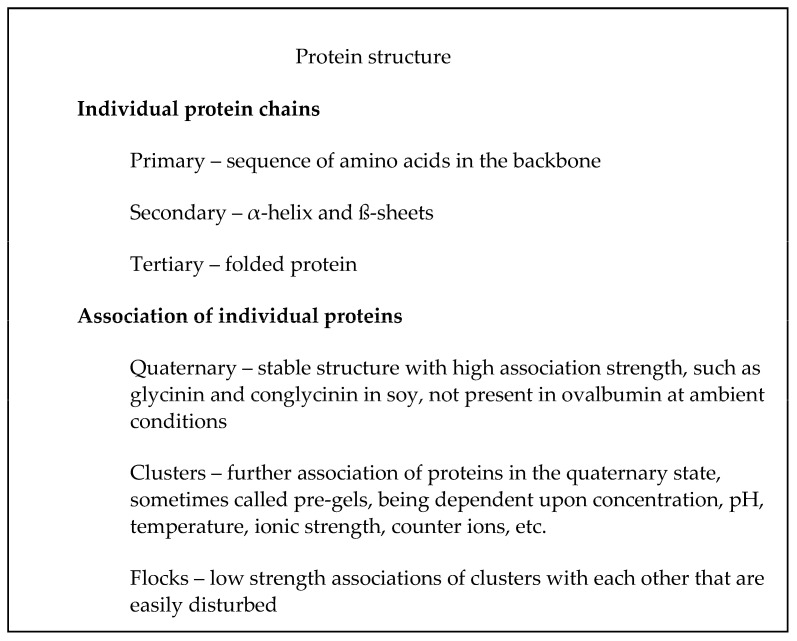
Levels of protein structure from the strong amino acid backbone to the weak flocks.

## Data Availability

Not Applicable.

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
