# Peer review of "Use of Dynamic Shear Rheology to Understand Soy Protein Dispersion Properties"

_polymers, 2022, doi:10.3390/polym14245490_

Round 1

Reviewer 1 Report

The article by Frihart C.R. and Gargulak M. examines the nonlinear viscoelasticity of aqueous dispersions of soy flour with different solubility, concentration, and presence of additives. Their interest in this research is due to the applicability of the studied systems as adhesives, whose viscosity and content of solids determines their applicability for wood gluing. The authors examine how various conditions, including the presence of a defoamer, a crosslinker, and a disulfide-bond breaker affect the viscoelastic characteristics of the dispersions, and briefly compare their systems with commercial adhesives. As a result, the authors find that soy flour-based adhesives break down under the action of shear in two stages and make some assumptions about the reasons. In general, the article is very detailed and clearly written and can be published in Polymers after revision.

Specific comments are as follows.

Lines 5-7: There is no postal data of the authors.

Line 249: “It is clear from Figures 5 and 6 that the soy dispersions have considerable non-Newtonian properties.” This is not really a correct statement. The non-Newtonian behavior is the presence of a dependence of steady-state viscosity on shear rate or shear stress, whereas the authors measure the storage and loss moduli under non-steady-state conditions and at large amplitudes. These are not the same thing. By way of example, if we take a melt of a monodisperse polymer, it is a Newtonian fluid because its viscosity does not depend on the shear rate. However, if we measure the amplitude dependences of the moduli of this melt, the storage and loss moduli will decrease beyond the linear region. In other words, the mention of non-Newtonian properties without considering flow curves is better to remove.

Figure 7: Please indicate in the caption (or in the legend) the content of solids for each sample.

Line 277: “With neither the insoluble nor the purified protein fractions showing a second shear thinning transition, a possible explanation is that one of the transitions may be due to protein-carbohydrate interactions [26].” I do not agree, as it seems that the situation is much simpler. The first (low-amplitude) transition for soy flour dispersion is the destruction of the fragile interparticle network from insoluble carb. This network is brittle and fractures at a strain amplitude of about 0.1-1% (such a level of critical strain is typical for gels made of suspended dispersed particles, see, e.g., 10.3933/ApplRheol-24-13653). The second transition is the disentanglement of macromolecules of soluble proteins. This transition is observed at a strain of the order of 25-100% (such a level of critical strain is typical for polymer melts and entangled polymer solutions, see, e.g., 10.1134/S0965545X14010039). In other words, these two transitions are a combination of one transition from the interacting insoluble particles and a second transition from the entangled dissolved macromolecules. I think if the authors present Figure 7 in the "moduli vs. strain" coordinates, they can see the coincidence of the magnitudes of strain of the two transitions in the soy flour dispersion with the characteristic critical strains in the native soy protein isolate and soy insoluble fraction.

Line 312: “KDa” -> “kDa”.

Lines 489-494, 516-518: An alternative might be: The first transition is the breaking of interparticle interactions between insolubles, while the second transition is the macromolecular disentanglement of solubles. In other words, the explanation lies in the fact that some of the soy flour dissolves and forms entanglements, and some does not dissolve and interacts as particles.

Lines 519-520: “For research articles with several authors, a short paragraph specifying their individual contributions must be provided. The following statements should be used”. These lines need to be deleted.

Line 525: “Please add:”. This is unnecessary.

Author Response

We appreciate the reviewer's careful and thoughtful review of the paper.

Reviewer 2 Report

The dynamic analysis used in order to obtain and interpreted the rheological data is correctly and clearly shown.

By dynamic shear rheology and complementary analysis was easy to understand the soy dispersions properties, and the conclusions of your work.

Author Response

We appreciate your review of this manuscript and have made some clarifications on the basis of another reviewer.